# The Effect of Super-Repressor IkB-Loaded Exosomes (Exo-srIκBs) in Chronic Post-Ischemia Pain (CPIP) Models

**DOI:** 10.3390/pharmaceutics15020553

**Published:** 2023-02-07

**Authors:** Ji Seon Chae, Hyunju Park, So-Hee Ahn, Eun-Chong Han, Yoonjin Lee, Youn Jin Kim, Eun-Jin Ahn, Hye-Won Oh, Hyun Jung Lee, Chulhee Choi, Youn-Hee Choi, Won-joong Kim

**Affiliations:** 1Department of Anesthesiology and Pain Medicine, College of Medicine, Ewha Womans University, Seoul 07804, Republic of Korea; 2Department of Physiology, Inflammation-Cancer Microenvironment Research Center, College of Medicine, Ewha Womans University, Seoul 07804, Republic of Korea; 3ILIAS Innovation Center, ILIAS Biologics Inc., Daejeon 34014, Republic of Korea; 4Department of Anesthesiology and Pain Medicine, College of Medicine, Chung-Ang University, Seoul 06974, Republic of Korea

**Keywords:** exosomes, exosome loaded with super-repressor IκB, chronic post-ischemic pain model, complex regional pain syndrome (CRPS), allodynia, NFκB, inflammation, neuropathic pain

## Abstract

Complex regional pain syndrome (CRPS) is a condition associated with neuropathic pain that causes significant impairment of daily activities and functioning. Nuclear factor kappa B (NFκB) is thought to play an important role in the mechanism of CRPS. Recently, exosomes loaded with super-repressor inhibitory kappa B (Exo-srIκB, IκB; inhibitor of NFκB) have been shown to have potential anti-inflammatory effects in various inflammatory disease models. We investigated the therapeutic effect of Exo-srIκB on a rodent model with chronic post-ischemia pain (CPIP), a representative animal model of Type I CRPS. After intraperitoneal injection of a vehicle, Exo-srIκB, and pregabalin, the paw withdrawal threshold (PWT) was evaluated up to 48 h. Administration of Exo-srIκB increased PWT compared to the vehicle and pregabalin, and the relative densities of p-IκB and IκB showed significant changes compared to the vehicle 24 h after Exo-srIκB injection. The levels of several cytokines and chemokines were reduced by the administration of Exo-srIκB in mice with CPIP. In conclusion, our results showed more specifically the role of NFκB in the pathogenesis of CRPS and provided a theoretical background for novel treatment options for CRPS.

## 1. Introduction

Complex regional pain syndrome (CRPS) is a condition associated with neuropathic pain and is characterized by intractable pain, multiple system dysfunction, and motor and autonomic dysfunction [1]. CRPS can be classified into two subtypes based on the presence (type II) or absence (type I) of direct nerve injuries [2,3,4]. There are several medical treatments available for CRPS, such as prophylactic vitamin C, narcotics, anticonvulsants, such as gabapentin or pregabalin, tricyclic antidepressants, and ketamine. However, in many cases these medications do not often have high efficacy. These therapeutic limitations are partly attributed to the insufficient evidence on the pathogenesis of CRPS [1,2,3,4].

The transcription factor nuclear factor kappa B (NFκB) has received considerable scientific attention over the past few years as a key factor in inflammation, apoptosis, and neuronal–glial interactions [5]. Excessive NFκB activity is associated with the pathogenesis of several chronic inflammatory disorders [6]. NFκB resides in the cytosol of different cell types and can be activated by various triggers [1]. Upon the activation of NFκB, IκB is phosphorylated by IκB kinases (IKK) at two conserved serine residues in the N-terminus. Following phosphorylation, IκB is ubiquitinated and then degraded by the 26 S proteasome. The IκB degradation is the key stage in NFκB activation, where it leads to its rapid nuclear translocation [7].

A previously described automated analysis of the literature has revealed that NFκB is involved in the pathogenesis of CRPS [1]. The affected limbs of patients with CRPS showed signs of chronic ischemia, which can trigger NFκB activation by the formation of reactive oxygen species and peroxynitrite. The biological fluids in the blister and the spinal cord contain high levels of inflammatory mediators, including tumor necrosis factor alpha, interleukin-1 (IL-1), and IL-6, which can activate or be induced by NFκB. Moreover, the abnormally expressed neuropeptides in CRPS, such as calcitonin gene-related protein and substance P, interact with NFκB [3,4,8].

Coderre et al. introduced the chronic post-ischemia pain (CPIP) model induced by ischemia and reperfusion (I/R) injury of the hind paw of rats [9]. Under general anesthesia, rats with CPIP undergo placement of a tight-fitting tourniquet around their hind paw for a 3 h period followed by reperfusion. Chronic ipsilateral and more sporadic contralateral, mechanical, and cold allodynia were demonstrated to follow I/R injury for at least 4 weeks [9]. The CPIP model shows several features that resemble CRPS, including edema, hyperemia, and the development of mechanical and cold allodynia without direct nerve injuries. Accordingly, the CPIP model has been proposed as an animal model for type I CRPS [9,10,11].

Exosomes have been recognized as potent therapeutics or drug delivery vehicles for transferring various materials, including proteins and regulatory genes, to target cells. The nonimmunogenic nature of these nanovesicles enables them to protect their cargo from serum proteases and immune responses [12,13,14,15]. A recent, novel, and optogenetically engineered exosome technology using “exosomes for protein loading via optically reversible protein–protein interactions” (EXPLOR) was developed [16]. This EXPLOR technology has been adopted in inflammatory diseases such as sepsis, acute kidney injury, and preterm birth [13,17,18]. Engineered exosome loaded with super-repressor IκB (Exo-srIκB), which is a non-degradable form of IκB that prevents the nuclear translocation of NFκB, can be proposed as a potential therapeutic approach for CRPS [13,17,18].

The aim of the present study was to examine whether the intraperitoneal administration of Exo-srIκB can reduce the mechanical allodynia in the CPIP model via changing the levels of p-IκB, IκB, cytokines, and chemokine-related inflammation in the affected paws.

## 2. Materials and Methods

### 2.1. Study Design

First, in the process of making the CPIP model, the mechanical allodynia and rotarod tests were performed for 48 h. A total of 30 min after the CPIP model was made, the mechanical allodynia and rotarod tests were performed again for four groups administered intraperitoneally with 1 × 10^9^ particle number (pn) per 0.1 mL of the vehicle, 1 × 10^9^ pn per 0.1 mL Exo-srIκB, 1 × 10^10^ pn per 0.1 mL Exo-srIκB (10 times higher concentration), and 0.1mL of 30 mg/kg pregabalin (positive control) (8 mice per each group). After the drug injection, the mechanical allodynia and rotarod tests were performed for 48 h again (Figure 1A). The sham model applied a pre-cut O-ring to the left paw in the same conditions as the CPIP model. We collected paw tissues affected from the sham and CPIP models (8 mice per each group) (Figure 1B). For the four groups, tissues of paws affected were obtained after 24 and 48 h of intraperitoneal injection for Western blot (6 mice representing each group) (Figure 1C). Cytokine and chemokine arrays were performed with tissues of paws affected of the sham, the vehicle, Exo-srIκB, and pregabalin groups after 24 h of injection (6 mice representing each group) (Figure 1D).

### 2.2. Animals

This study was approved by the Institutional Animal Care and Use Committee of the Ewha University College of Medicine (protocol code No: EWHA MEDIACUC 22-038). Male C57BL/6 mice (20–25 g) were used in the experiments. They were allowed to drink and eat freely, and a 12/12 h light/dark cycle was used. All animals were left for 7 days to adapt to their environment.

### 2.3. CPIP Model Making

The CPIP model was induced using the method introduced by Coderre et al. [9]. General anesthesia was performed using 1.5% isoflurane and 100% O_2_ in all the groups. In the CPIP group, an O-ring with a 5/64-inch internal diameter that matched the size of the mouse’s hind limb was placed on the upper part of the left ankle (immediately above the medial malleolus) for 3 h. The hind paw showed evidence of hypoxia (cold and cyanotic) when the O-ring was placed (Figure 2A). The tied O-ring was removed 3 h after ischemia to induce reperfusion, and the mice were awakened from the anesthesia. After reperfusion, there was a period of hyperemia (Figure 2B). The edema persisted for 3 days with a gradual return to baseline. In the sham group, precut O-rings of the same size were applied to prevent a tightening force.

### 2.4. Exosome: Exo-srIκB

As described in previous studies, the generation and isolation of Exo-srIκB was achieved through the natural biogenesis process of exosomes and reversible protein–protein interactions regulated by optogenetics [13,16]. The characterization of Exo-srIκB measured exosome particle number per volume using nanoparticle tracking analysis (NTA), and the presence of positive/negative exosome markers using Western blotting. We confirmed exosome positive and negative markers through Western blotting. Exo-srIκB has loaded target proteins, including srIκB, CRY2, and CD9 (CD9-CIBN), and positive markers (endogenous CD9, CD81, TSG101, Alix, and GAPDH).

### 2.5. Drugs

In our study, we injected the following drugs in the mice intraperitoneally; a vehicle (1 × 10^9^ pn/mL), Exo-srIκB (11 × 10^9^ pn/mL), 10× Exo-srIκB (11 × 10^10^ pn/mL), and pregabalin (30 mg/kg) using 0.1 mL each. As a negative control, we used non-engineered exosomes derived from HEK293 F cells. Pregabalin was used as a positive control because it has an anti-inflammatory mechanism associated with NFκB through regulating the release of sensory neuropeptides [19]. The dose of Exo-srIκB was based on the dose obtained from another study in which the same substance was injected intraperitoneally to obtain an anti-inflammatory effect [13]. The concentration of pregabalin was also based on a study on neuropathic pain in mice [20].

### 2.6. Mechanical Allodynia

To confirm the establishment of the CPIP mice model, mechanical allodynia was assessed by measuring the bilateral paw withdrawal thresholds (PWTs) to von Frey filaments just before applying the O-ring, and at 30, 60, 90, 120 min, 24, and 48 h after the application of the O-ring, referring to the experimental method reported by Kim et al. [10]. To stimulate the plantar surface, a mouse was placed in a Plexiglas cage with a wire grid bottom. The wire grid floor was unexposed to iron and coated with plastic, so it was possible to minimize being too cold or irradiated to animals. Furthermore, the mouse was acclimatized to the environment for approximately 30 min before the experiment. A vertical force was applied to the paw of the mouse for 3 s with a von Frey filament, such that the filament bent in the midplantar area, and the avoidance response was then evaluated. For each filament measurement, an interval of at least 30 s was given. Nine filaments were used, with weights ranging between 2.44 and 4.56 (0.04–4 g). The simplified up-down method used by Bonin et al. [21] was used to check the reflex four additional times, when the mouse began to show or discontinued showing an avoidance response. A 50% response threshold was measured based on the reflex patterns and log-value of the von Frey filament. A log scale in grams value was used for the PWTs [22,23].

### 2.7. Tissue Sampling and Preparation

Mice were euthanized by decapitation under general anesthesia with isoflurane. Muscle of the superficial plantar layer were immediately obtained and frozen in isopentane, kept on dry ice, and stored at −80 °C until processing. Samples were thawed at 4 °C and homogenized by sonification in 12 mL/mg tissue of RIPA buffer containing 50 mM Tris-HCl, 150 mM NaCl, 1 mM EDTA, 1% Igepal (Sigma–Aldrich, St. Louis, MO, USA), 1% Sodium deoxy-cholate, and 1% SDS (pH 7.4), and a 1% protease inhibitor cocktail was added to the buffer (Sigma–Aldrich). Homogenates of tissue were centrifuged at 3000× *g* for 10 min. Proteins were quantified using the bicinchoninic acid (BCA) assay. Extracts were stored at −80 °C until further experiment.

### 2.8. Western Blot

The paw tissue homogenates (35 µg protein/lane) were separated using an 8–10% SDS-PAGE gel and transferred to nitrocellulose membrane by electrophoresis. The mem-branes were probed with specific antibodies against p-IκB, IκB, and tubulin (1:1000 dilution) for 20 h, followed by the addition of secondary antibodies (1:1000) for 30 min. Detection was performed using an enhanced chemiluminescence detection kit (Thermo Scientific, Waltham, MA, USA).

### 2.9. Rotarod Test

To rule out any sedative effects or motor disturbances elicited by Exo-srIκB, the rotarod test was conducted [20]. Animals were habituated to the rotarod instrument for two consecutive days at low-speed rotation (5 r/min) for 600 s each day before actual measurement. Mice that could not stay on the rod for 600 s were excluded from the experiment. During the experiment, the animals had undergone the test in three accelerating trials of 300 s with the rotarod speed increasing from 5 to 40 r/min over the first 120 s. There was an intertrial interval of at least 20 min for each mouse. The falling latency of mice was checked for each trial with a cutoff time of 300 s.

### 2.10. Cytokine and Chemokine Array

Cytokine and chemokine arrays were performed with homogenized paws of the sham, vehicle, Exo-srIκB, and pregabalin groups (6 mice representing each group) using Proteome Profiler Mouse Cytokine Array Kit ARY028 and Chemokine Array Kit ARY020, respectively (R&D Systems, Minneapolis, MN, USA). The sample/antibody mixture was then added onto the blocked membrane which contained different capture antibodies. After washing, the membrane was incubated with diluted streptavidin-horseradish peroxidase. A Chemi Reagent Mix was added and the membrane was then exposed to X-ray film.

### 2.11. Statistical Analyses

Mean ± standard error of the mean (SEM) was calculated in all bar graphs from independent experiments. The statistical significance was analyzed by ANOVA and Student’s t-test using SPSS software version 23.0 (SPSS Inc., Chicago, IL, USA). Statistical significance of the data was set at a *p*-value of <0.05.

## 3. Results

### 3.1. Characterization of Exo- srIκB

The production of Exo-Naïve (non-engineered exosomes, vehicle) and Exo- srIκB (engineered exosomes) have been previously described [16]. The nanoparticle tracking analysis measured exosome size and concentration (NTA, Figure 3A). We confirmed exosome positive and negative markers through Western blotting. Exo-Naïve (vehicle) did not express target protein srIκB, CRY2, and CD9 (CD9-CIBN), but expressed positive markers (Figure 3B). Exosomes did not detect the expression of cell organelle markers, including GM130, Lamin B1, Prohibitin, and Calnexin (Figure 3C).

### 3.2. CPIP Model and NFκB

The PWTs of both hind limbs were significantly decreased compared to the sham after 120 min of I/R injury (*p* < 0.05). Furthermore, the PWT of the ipsilateral hind limb at 48 h after I/R injury was significantly different compared to the contralateral hind limb (*p* < 0.05) (Figure 4A). The relative density of p-IκB in the CPIP group significantly increased compared with that in the sham group; while the relative density of IκB in the CPIP group decreased compared with that in the sham group (*p* < 0.001 and *p* < 0.01, respectively) (Figure 4B).

### 3.3. Anti-Allodynic Efficacy of Exo-srIκB

The PWTs of the ipsilateral hind paw significantly decreased compared to baseline in all the groups, but significantly increased in Exo-srIκB, and 10× Exo-srIκB groups at 60 min, 120 min, 24 h, and 48 h after I/R injury compared to those in the vehicle groups. The PWTs in the Exo-srIκB and 10x Exo-srIκB groups were significantly increased compared to pregabalin at 48 h (*p* < 0.05) (Figure 5A).

The PWTs of the contralateral hind paw significantly decreased compared to the baseline in all groups, but did not show significant differences between the groups (Figure 5B).

The rotarod test showed that there were different levels of motor coordination in all the groups (Figure 5C).

### 3.4. P-IκB and IκB Levels after Exo-srIκB Injection in Western Blot

The relative density of p-IκB in the Exo-srIκB group decreased significantly compared to the vehicle group (*p* < 0.05). The relative density of IκB in the Exo-srIκB and 10× Exo-srIκB groups increased significantly compared to the vehicle group (Figure 6A) (*p* < 0.05 and *p* < 0.001, respectively). The relative densities of p-IκB and IκB in the paw showed no significant changes in all the groups 48 h after injection (Figure 6B).

### 3.5. Reductions in the Levels of Cytokines and Chemokines after Injection of Exo-srIκB

The levels of endoglin, myeloperoxidase, osteopontin, pentraxin 3, and serpin E1/PAI-1 were significantly enhanced in the vehicle group compared with those in the sham group, showing the infiltration of inflammatory cells in the paws. However, the protein expression of endoglin, myeloperoxidase, osteopontin, pentraxin 3, and serpin E1/PAI-1 in the paws was reduced remarkably when treated with Exo-srIκB (* *p* < 0.05, ** *p* <0.01, *** *p* < 0.001 vs. vehicle). The expression levels of CCL21, CXCL2, IGFBP-5, and IL-33 showed an increase significantly in the Exo-srIκB injection group compared to those in the vehicle group (* *p* < 0.05, ** *p* < 0.01, *** *p* < 0.001 vs. Exo-srIκB) (Figure 7).

The levels of C10, the complement component C5/C5a, MCP-2, MIP-1 γ, IL-16, MCP-5, and SDF-1 were significantly enhanced in the vehicle group compared with those in the sham group. The expression of C10, the complement component C5/C5a, MCP-2, MIP-1 γ, IL-16, MCP-5, and SDF-1 were reduced remarkably when treated with Exo-srIκB (* *p* < 0.05, ** *p* < 0.01, *** *p* < 0.001 vs. vehicle). Moreover, IL-16, MCP-5, and SDF-1 showed a decrease upon treatment with pregabalin (* *p* < 0.05, ** *p* < 0.01, *** *p* < 0.001 vs. vehicle) (Figure 8).

## 4. Discussion

Our findings showed that the PWTs of both hind limbs were significantly reduced compared to the baseline, and the relative densities of p-IκB and IκB were significantly altered in the CPIP mice compared to those in the sham group. The administration of Exo-srIκB increased the PWTs of the affected hind limbs compared to the vehicle and pregabalin groups, and the relative densities of p-IκB and IκB showed a significant change compared to the vehicle group 24 h after drug injection.

There are several studies on the involvement of NFκB in the CPIP model. De Mos et al. [3] assessed the anti-allodynic effect of pyrrolidine dithiocarbamate (PDTC), a NFκB inhibitor, and measured NFκB levels via enzyme-linked immunosorbent assay. They reported that systemic PDTC treatment relieved mechanical allodynia in a dose-dependent manner. They also showed that the levels of NFκB were elevated in the muscles of CPIP rats compared to sham rats 2 and 48 h after I/R injury. Ross-Huot et al. [24] also demonstrated that the levels of NFκB in the ipsilateral hind paw muscles were higher in the relatively hyperglycemic groups 2 days post-I/R injury. Accordingly, the administration of SN50, a cell-permeable synthetic peptide which inhibits the translocation of the active NFκB complex into the nucleus [25,26], resulted in the reduction of mechanical allodynia. The reported findings are consistent with our results where the Exo-srIκB group showed an increase in the PWTs of the hind limbs compared to the vehicle group.

PDTC does not directly inhibit IκB phosphorylation, but rather inhibits the signaling required for IκB degradation via proteasome rats [4,8]. In addition, PDTC inhibits the oxidative stress in vivo and in vitro by increasing the antioxidant capacity. Therefore, PDTC can contribute to the inhibition of NFκB activation through multiple mechanisms [4]. Moreover, SN50 blocks the nuclear import of NFκB and other transcription factors, such as activator protein 1, signal transducers and activator of transcription, and nuclear factor of activated T-cells [27,28]. However, Exo-srIκB directly blocks the nuclear translocation of NFκB only without any additional mechanisms unlike the other substances. Accordingly, Exo-srIκB more selectively inhibits the expression of specific NFκB target genes.

Forty-eight hours after drug injection, the effect of Exo-srIκB disappeared as observed in the results of the Western blot, but the differences of the PWT were more pronounced in the Exo-srIκB group compared to the vehicle group. A previous study that injected GABA intrathecally for animal models with neuropathic pain after nerve injury predicted that these sensory behaviors could be temporarily or permanently returned to the pre-pain condition, and experiments showed that a single injection of GABA could alleviate neuropathic pain. However, this effect did not appear after 2–3 weeks of nerve damage, and they confirmed that proper early treatment resulted in persistent alleviation of pain [29]. The previous study that confirmed whether organ damage and mortality were recovered by injecting Exo-srIκB in sepsis animal models also reported that a single injection of Exo-srIκB intraperitoneally inhibited the secretion of proinflammatory cytokines, preventing overwhelming inflammatory response, and thus alleviating the mortality and systemic inflammation of septic mouse models [13]. This study also seems to have resulted in this persistent effect by preventing excessive inflammatory action by injecting the exosome at an appropriate time in the early stage of neuropathic pain.

In addition to the CPIP model, there are several studies on the efficacy of Exo-srIκB [13,17,18]. The administrations of Exo-srIκB in the septic mouse model alleviated mortality and the systemic inflammatory response [13]. Moreover, Exo-srIκB treatment during lipopolysaccharide-induced preterm birth prolonged gestation and reduced maternal inflammation [17]. Furthermore, the systemic delivery of Exo-srIĸB decreased NFĸB activity in the post-ischemic kidneys and reduced apoptosis. The post-ischemic kidneys showed decreased gene expression of the pro-inflammatory cytokines and adhesion molecules with Exo-srIĸB treatment compared with the control [18].

In this study, almost all the cytokines and chemokines were identified in the paw tissues of mice, and showed a significant difference in all the groups. In the CPIP model, characteristic cytokines and chemokines related to inflammation were observed after reperfusion in the paw tissues. The cytokines that showed a distinct difference between the vehicle group and the Exo-srIκB group include myeloperoxidase, osteopontin, pentraxin 3, serpin E1/PAI-1, and endoglin/CD105. The chemokines include C10, the complement component C5/C5a, monocyte chemoattractant protein-2, macrophage inflammatory protein-1γ, IL-16, MCP-5, and stromal cell-derived factor 1. Myeloperoxidase is an enzyme that is present in the granules of leucocytes, mainly neutrophils and macrophages, and is responsible for secreting hypochlorite, which is considered an inflammatory process marker [30]. Several other reports showed an increase in myeloperoxidase in CPIP, indicating the possibility that myeloperoxidase may play an important role in the inflammatory process of CRPS/CPIP [31,32,33]. Pentraxin 3, an inflammatory marker and a pattern recognition receptor, plays a crucial role in the exacerbation of inflammatory diseases [34]. Pentraxin 3 expression is controlled by various signaling pathways, such as NFκB, JNK, and PI3K/Akt. Pentraxin 3 has been identified as an inducible gene of IL-1 and TNF-α, which are widely related in the regulation of inflammatory diseases, such as inflammatory-related tumors and neuroinflammation [34]. Activation by the interaction of C5a and its receptor, C5aR1, triggers a cascade of events that are involved in the pathophysiology of peripheral neuropathy and painful neuro-inflammatory states [35,36]. MCP-2 and MIP-1 γ are chemokines associated with macrophage, which are associated with inflammation, and IL-16 is also a lymphocyte chemoattractant factor that interacts with CD4 and serves as a key mediator in inflammatory reactions [37,38,39,40,41]. To our knowledge, this is the first study to investigate the increased levels of cytokines and chemokines in response to neuropathic pain and NFκB through arrays and confirm the effectiveness of exosomes. It is necessary to clarify the relationship between CPIP and some cytokines and chemokines. Thus, additional cytokine or chemokine knock-out experiments need to be conducted on animals.

Pregabalin, which was used as a positive control in our study, has also been shown to possess anti-oxidative, anti-TNF-alpha, and anti-inflammatory actions [19]. It modulates the release of sensory neuropeptides, such as substance P and CGRP, in inflammation-induced spinal cord sensitization. Furthermore, it inhibits NFκB activation in both neuronal and glial cell lines by suppressing substance P and other inflammatory neuropeptides [19]. Accordingly, our results showed the decreased relative density of p-IκB and the increased relative density of IκB in the pregabalin group compared to the vehicle group 24 h after pregabalin injection. In addition, the cytokine and chemokine arrays showed that some proteins in the Exo-srIκB and pregabalin groups had the same patterns.

In conclusion, intraperitoneal administration of Exo-srIκB reduced mechanical allodynia through NFκB inhibition, and a number of cytokines and chemokines were found to be involved in the CPIP mice. Our results showed more specifically the role of NFκB in the pathogenesis of CRPS and provided a theoretical background for new treatment options for CRPS.

## Figures and Tables

**Figure 1 pharmaceutics-15-00553-f001:**
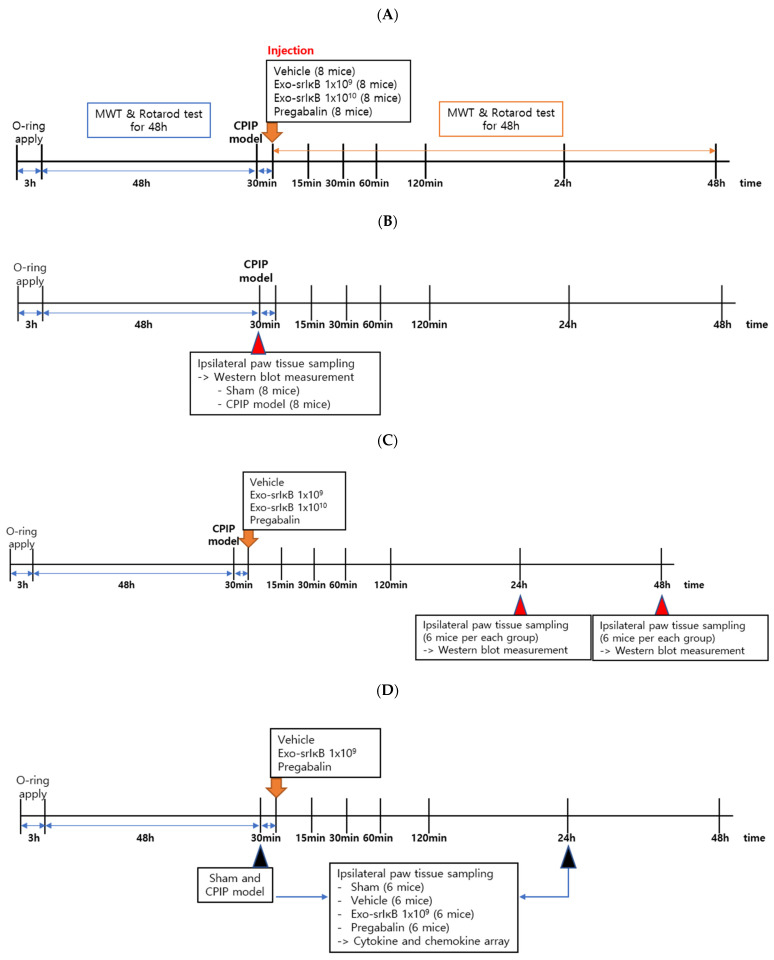
Experimental protocol. (**A**) Mechanical allodynia and rotarod tests; (**B**) Western blot measurement for sham and CPIP models; (**C**) Western blot measurement for four groups at 24 h and Western blot measurement for four groups at 48 h after drug injection; (**D**) cytokine and chemokine measurement for four groups at 24 h after drug injection.

**Figure 2 pharmaceutics-15-00553-f002:**
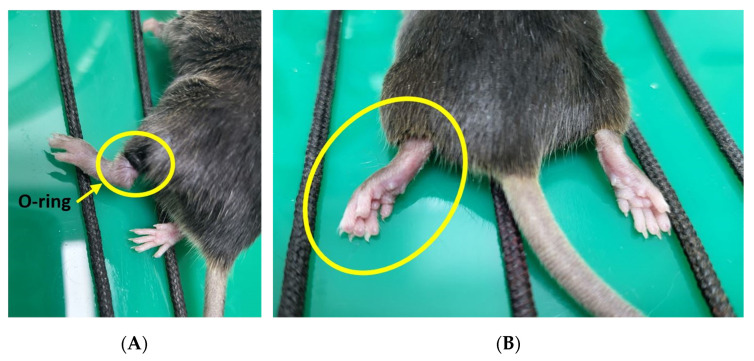
Chronic post-ischemia pain model. (**A**) An O-ring was placed on the upper part of the left ankle for 3 h; (**B**) the tied O-ring was removed after 3 h of ischemia to induce reperfusion. There was a period of hyperemia after perfusion.

**Figure 3 pharmaceutics-15-00553-f003:**
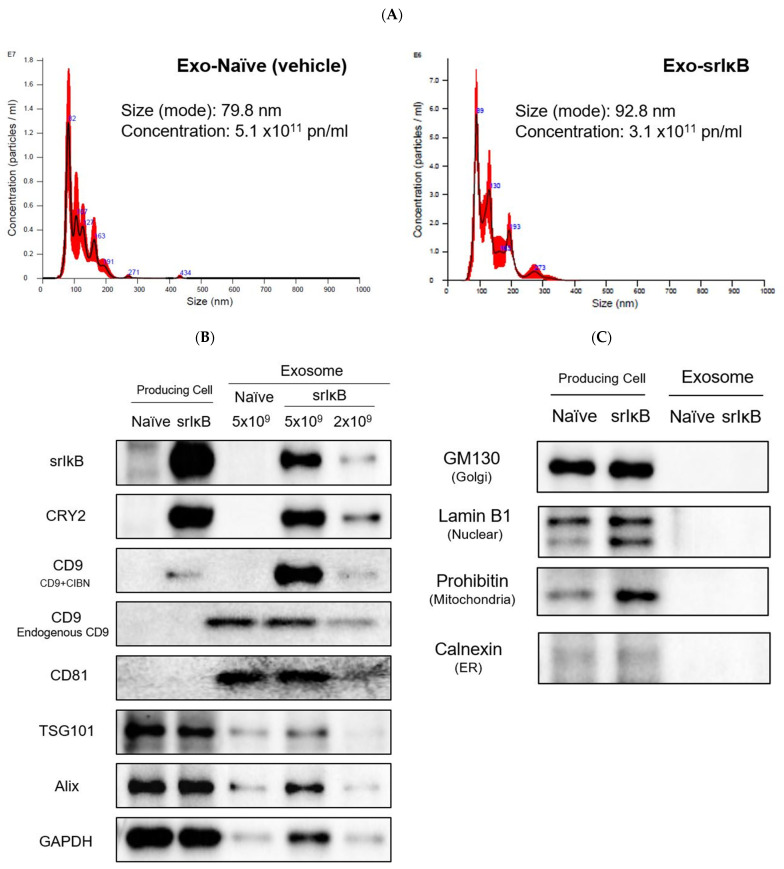
Characterization of Exo-srIκB. Exo-Naïve (vehicle) is non-engineered exosome and Exo-srIκB is engineered exosome. A vehicle used as negative control of Exo-srIκB. (**A**) Size distribution and concentration of the vehicle (left) and Exo-srIκB (right) were determined by a Nanosight NS 300 (Malvern Panalytical, United Kingdom); (**B**) Western blot at producing cell and exosome to confirm the expression of target protein (srIκB, CRY2, and CD9 (CD9-CIBN)), exosome-positive markers (endogenous CD9, CD81, TSG101, Alix, and GAPDH) (The original western blot images are in Appendix A); and (**C**) exosome-negative markers (cell organelle markers; GM130 (Golgi), Lamin B1 (Nuclear), Prohibitin (mitocondria), and Calnexin (ER)) (The original western blot images are in Appendix A).

**Figure 4 pharmaceutics-15-00553-f004:**
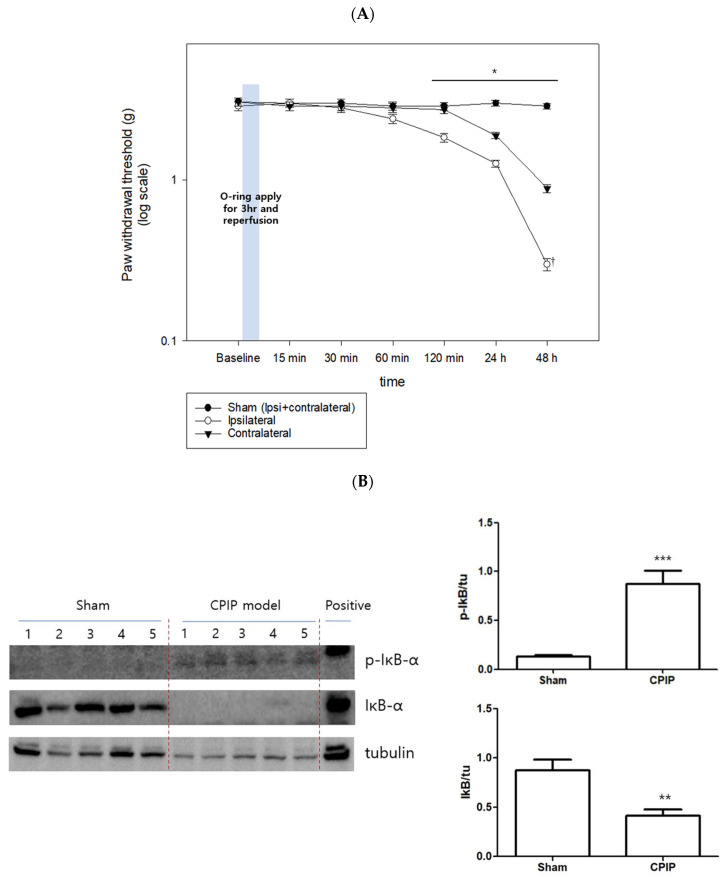
CPIP model and NFκB. (**A**) Paw withdrawal thresholds. * *p* < 0.05 vs. sham, ^†^ *p* < 0.05 vs. contralateral; (**B**) Western blot data showing the relative densities of p-IκB and IκB to tubulin at 48 h after ischemia and reperfusion injury. ** *p* < 0.01, *** *p* < 0.001 vs. sham. IκB: inhibitory kappa B, p-IκB: phospho inhibitory kapp B (The original western blot images are in Appendix A).

**Figure 5 pharmaceutics-15-00553-f005:**
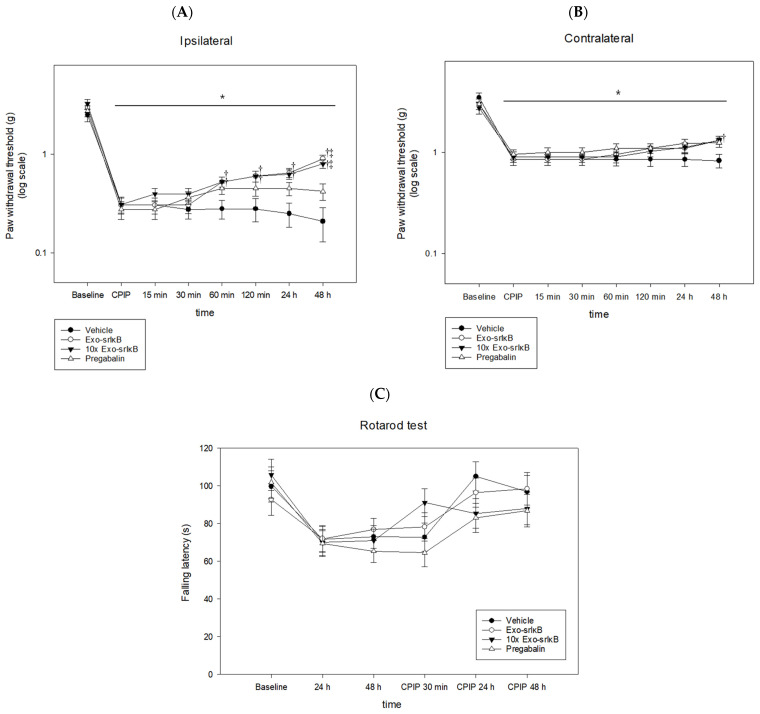
Antiallodynic effects of Exo-srIκB. (**A**) Ipsilateral paw withdrawal threshold; (**B**) contralateral paw withdrawal threshold; (**C**) rotarod test. * *p* < 0.05 vs. baseline, ^†^ *p* < 0.05 vs. vehicle, ^‡^ *p* < 0.05 vs. pregabalin.

**Figure 6 pharmaceutics-15-00553-f006:**
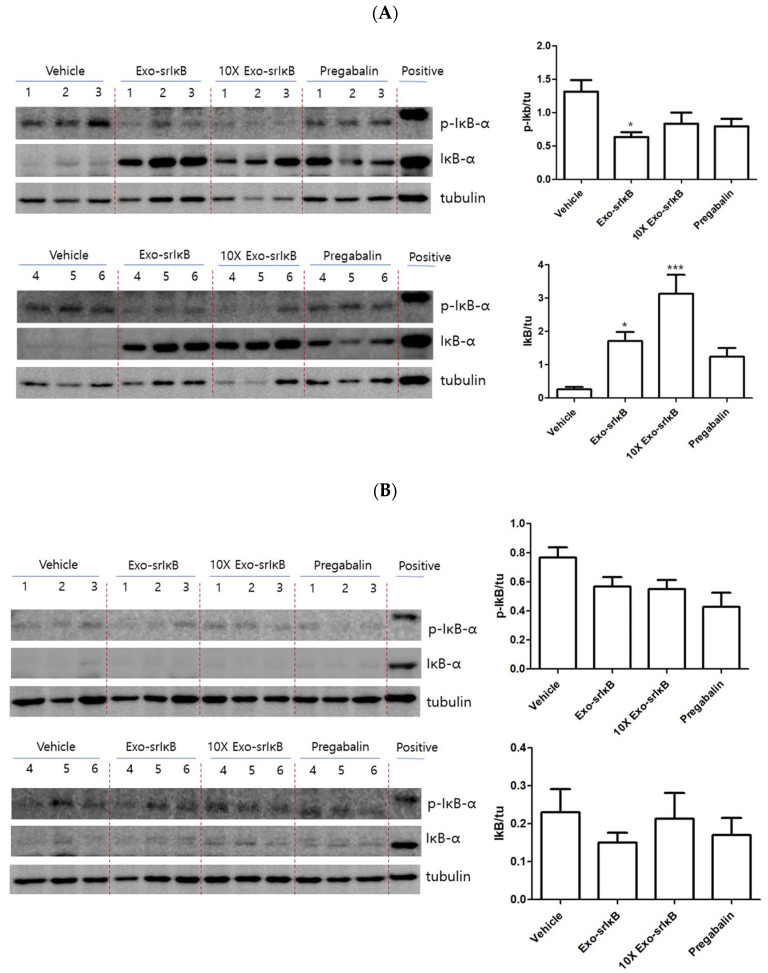
Western blot after drug injection. (**A**) Western blot at 24 h after drug injection. Western blot data showing the relative densities of p-IκB and IκB to tubulin at 24 h after drug injections (The original western blot images are in Appendix A); (**B**) Western blot at 48 h after injection. Western blot data showing the relative densities of p-IκB and IκB to tubulin at 48 h after drug injections. * *p* < 0.05, *** *p* < 0.001 vs. vehicle (The original western blot images are in Appendix A).

**Figure 7 pharmaceutics-15-00553-f007:**
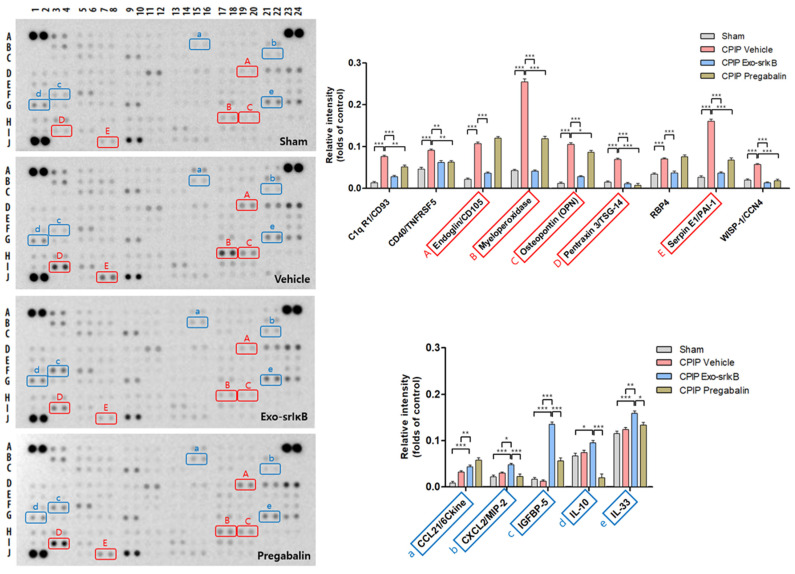
Cytokine array in the paw 24 h after drug injection. Cytokine profiles and plots of endoglin, myeloperoxidase (MPO), osteopontin (OPN), pentraxin 3, and serpin E1/PAI-1. * *p* < 0.05, ** *p* < 0.01, *** *p* < 0.001 vs. CPIP vehicle. Cytokine profiles and plots of CCL21, CXCL2, IGFBP-5, IL-10, IL-33, and LDL R s. * *p* < 0.05, ** *p* < 0.01, *** *p* < 0.001 vs. CPIP Exo-srIκB (The original Cytokine array images are in Appendix A).

**Figure 8 pharmaceutics-15-00553-f008:**
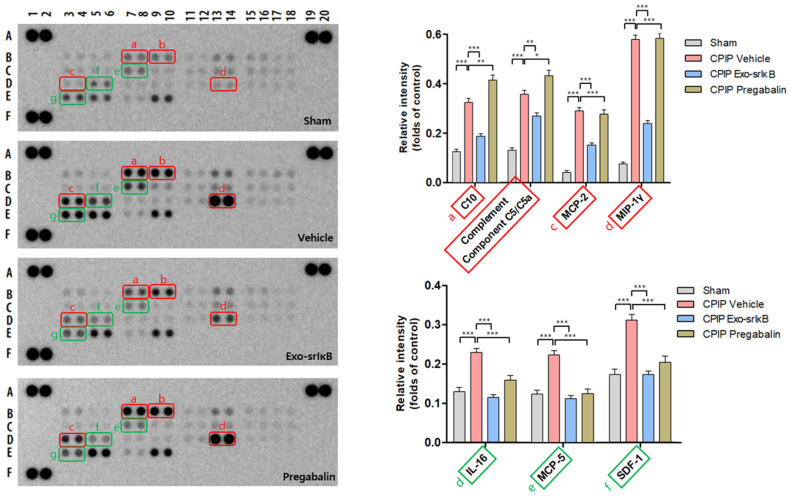
Chemokine array in the paw 24 h after drug injection. Chemokine profiles and plots of C10, the complement component C5/C5a, MCP-2, MIP-1 γ, IL-16, MCP-5, and SDF-1. * *p* < 0.05, ** *p* < 0.01, *** *p* < 0.001 vs. CPIP vehicle (The original Chemokine array images are in Appendix A).

## Data Availability

Not applicable.

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
