# Peer review of "The Effect of Super-Repressor IkB-Loaded Exosomes (Exo-srIκBs) in Chronic Post-Ischemia Pain (CPIP) Models"

_pharmaceutics, 2023, doi:10.3390/pharmaceutics15020553_

Round 1
Reviewer 1 Report
study design line number 87&90 mentioned 8 animals per group but the institutional animal ethical committee sanctioned only six animals per group why this difference
How sedative effect can be determine by rotarod test ? only motor impairment can be measured .The reference you mentioned also said the same
In figure 5 C rotarod test p values were not indicated in the graph
in figure 6 B statistical expression of P value was not indicated
In figure 7 statistical values of reductions in the levels of cytokines and chemokines after injection of Exo-srIκB was not mentioned
In line 317 the author mentioned about the hyperglycemic groups of animals but in the methodology and study design this was not mentioned
Author Response
We sincerely thank you for taking your precious time to review our paper. We have made some corrections to the manuscript after going over the comments. We have indicated the modifications made to the original document using red text.

Reviewer 2 Report
Complex regional pain syndrome (CPRS) is a refractory state characterized by pain and multiple system dysfunction, as represented by sympathetic nerve disorder. Whereas a number of treatments are available in clinics, none of these treatments exert a satisfactory effect. The authors investigated the treatment effect of suppressing nuclear factor kappa B (NFκB) using supper-repressor IKB-loaded exosomes (Exo-srIκB) in chronic post-ischemia pain model for CRPS. They showed that Exo-srIκB ameliorated the mechanical allodynia by changing the levels of p-IκB, IκB, cytokines, and chemokine-related inflammation. Moreover, the authors demonstrated that the pregabalin treatment applied as the positive control also showed similar results. Because CRPS is highly resistant to available treatments, developing a new treatment strategy is essential. On this ground, the reviewer considers the present study possesses high priority for publication. While the experiment seems reasonable and most of the results are presented well, the following points should be revised.
Major concerns;
Materials and Methods;
As the authors described, the result from the von Frey monofilament test depends on the degree of acclimatization in the assessment setting. However, the present experimental timeline does not allow the exact duration of acclimatization in the assessment apparatus in which the sensory testing is performed in 15, 30, and 60min. Particularly, the reviewer concerns that the wire grid bottom for the von Frey test is sometimes too cold or irradiative to the animals and may affect the result.
Moreover, allodynia model animals sometimes present behavior to scratch a wire grid bottom. In such case, the result becomes unreliable.
2.2 Animals. While the authors used relatively few mice in each group, the reviewer was concerned that this small sample size had enough statistical power.
Results;
While the authors performed western blotting to show IκB phosphorylation and cytokine array as the biochemical assessment, the reviewer is wondering if these results are sufficient to show the suppression of CRPS.
The reviewer considers that other evidence to show the suppression of CRPS-like change in the histological assessments should be needed.
Minor concerns;
Abstract:
The relationship between NFκB and IκB is not presented in the abstract. The reviewer considers that a short clause connecting them will help attract potential readers.
Discussion;
“Therefore, our results could more specifically explain that NFκB signaling is involved in the mechanisms of CRPS.”(Lines 336-337)
The reviewer considers that the current experiment does not necessarily show that NFκB signaling is involved in the mechanisms of CRPS because the authors just showed the amelioration of allodynia and inflammation.
Author Response

(The authors gave the same response as above.)

Round 2
Reviewer 2 Report
The reviewer considers the revised manuscript has sufficient quality for publication.
Author Response
Thank you for your valuable comment. There has been an improvement in our study due to your comments.